# Resonant tunneling driven metal-insulator transition in double quantum-well structures of strongly correlated oxide

R. Yukawa [1,5], M. Kobayashi[1,5], T. Kanda [2,5], D. Shiga [1,2,5], K. Yoshimatsu [2], S. Ishibashi [3], M. Minohara [1], M. Kitamura[1], K. Horiba [1], A. F. Santander-Syro [4] & H. Kumigashira [1,2✉]

The metal-insulator transition (MIT), a fascinating phenomenon occurring in some strongly correlated materials, is of central interest in modern condensed-matter physics. Controlling the MIT by external stimuli is a key technological goal for applications in future electronic devices. However, the standard control by means of the field effect, which works extremely well for semiconductor transistors, faces severe difficulties when applied to the MIT. Hence, a radically different approach is needed. Here, we report an MIT induced by resonant tunneling (RT) in double quantum well (QW) structures of strongly correlated oxides. In our structures, two layers of the strongly correlated conductive oxide $SrVO_3$ (SVO) sandwich a barrier layer of the band insulator $SrTiO_3$. The top QW is a marginal Mott-insulating SVO layer, while the bottom QW is a metallic SVO layer. Angle-resolved photoemission spectroscopy experiments reveal that the top QW layer becomes metallized when the thickness of the tunneling barrier layer is reduced. An analysis based on band structure calculations indicates that RT between the quantized states of the double QW induces the MIT. Our work opens avenues for realizing the Mott-transistor based on the wave-function engineering of strongly correlated electrons.

[1] Photon Factory, Institute of Materials Structure Science, High Energy Accelerator Research Organization (KEK), Tsukuba 305–0801, Japan. [2] Institute of Multidisciplinary Research for Advanced Materials (IMRAM), Tohoku University, Sendai 980-8577, Japan. [3] Research Center for Computational Design of Advanced Functional Materials, National Institute of Advanced Industrial Science and Technology (AIST), Tsukuba, Ibaraki 305-8568, Japan. [4] Université Paris-Saclay, CNRS, Institut des Sciences Moléculaires d'Orsay, 91405 Orsay, France. [5] These authors contributed equally: R. Yukawa, M. Kobayashi, T. Kanda, D. Shiga. ✉email: kumigashira@tohoku.ac.jp

Controlling the quantum ground state of a system is essential for applications. The best-known example is semiconductor technology, where the state (conductive or not conductive) of a semiconductor is driven by the so-called field-effect transistor (FET). In the FET, the number of electric charge carriers is controlled by an external voltage[1].

Some strongly correlated electron materials naturally show a metal-to-insulator transition (MIT)[2,3]. It would be thus highly desirable to control such an MIT in the same way as it is done for semiconductors. Numerous efforts have been made to demonstrate the FET control of the MIT[4–8], as well as of rich quantum phase transitions in strongly correlated electron materials[9–15]. For example, field-effect control over the MIT has been successfully achieved in many scenarios using electrolyte gating in the electric double-layer transistor[7,8], although such gate-controlled MIT is challenging because of the possible electrochemical reaction in the ionic liquid gate[16,17]. Meanwhile, the FET control of other correlated devices has been also achieved for specific cases via back gating[18–20]. Nonetheless, these important achievements have no pretense to result in a practical transistor[6,7,21]. In fact, such approach faces fundamental difficulties. One is the insufficient carrier density that can be induced by the electric field to cause a filling-controlled MIT (Mott transistor operation)[4,5]. The other is the shortness of the Thomas–Fermi screening length due to $10^{22}$–$10^{23}$ cm$^{-3}$ mobile carriers, which limits the conductive area where MIT occurs[7,8]. Thus, realistic future applications of the Mott transistor call for a different principle of controlling the MIT[22].

Now, according to the Mott–Hubbard theory[23], the ground state in strongly correlated materials is described by the ratio of the Coulomb interaction ($U$) to the bandwidth ($W$)[2,3,23]. When $U < W$, the material is metallic, but it becomes insulating when $U > W$. Therefore, tuning the $U/W$ ratio by some external perturbation would control the MIT, and the practical realization of this idea has been one of the central goals in modern condensed matter physics[4–11].

Here we propose a new approach for the tuning of the $U/W$ ratio, hence the control of the MIT, using the resonant-tunneling (RT) effects[1] in double quantum well (QW) structures of strongly correlated oxides (Supplementary Note 1). The concept is schematically illustrated in Fig. 1. The double QW structure consists of two strongly correlated oxide layers and a barrier layer (insulator). The top QW is a "marginal" Mott insulator, i.e., a material in the insulating Mott phase but in close proximity ($U$ slightly larger than $W$; $U \gtrsim W$) to the metallic one, while the bottom QW is a correlated metal. In the marginal Mott-insulating QW, the quantized electron states are localized due to $U \gtrsim W$, leading to a Mott insulating state, but the QW exhibits a transition to a metallic state by applying a small external stimulus. If the RT occurs between the marginal Mott-insulating QW and metallic QW states, the QW states that are energetically close to each other are hybridized. In this situation, the resultant envelope wave functions extend over the whole double QW structure and consequently electrons in the top QW states may achieve additional spatial freedom through the RT effects, leading to the reduction in effective Coulomb potential. As a result, the marginal Mott insulator is expected to be metallized owing to $U$ becoming smaller than $W$ (Supplementary Note 2). Since the MIT is caused by the quantum tunneling phenomena, the MIT control avoids the fundamental problems of the FET approach[4–21], and is also advantageous over conventional filling (bandwidth) control by chemical doping (pressure) in the bulk[2].

## Results

### Design of double QW structures showing RT-driven MIT. To demonstrate the MIT driven by the RT effects, we fabricate

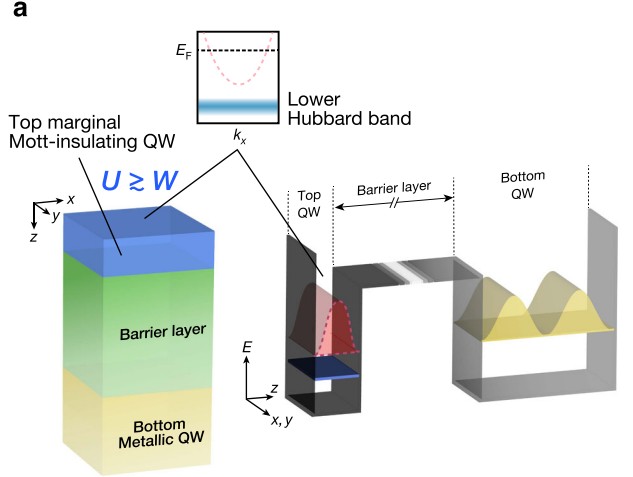

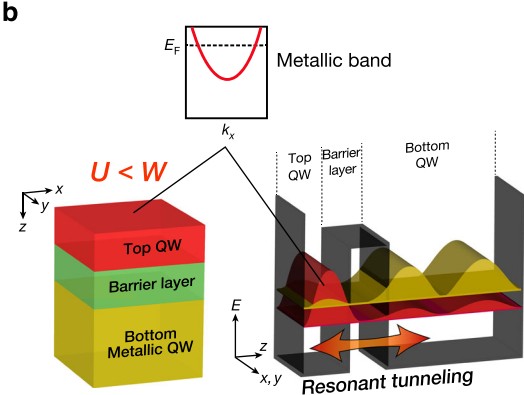

**Fig. 1 Schematic illustration of the metal–insulator transition induced by resonant tunneling effects. a** Before switching on the RT effect. The structure of a double QW consists of a marginal Mott-insulator QW layer/barrier layer/metallic QW layer. The marginal Mott-insulator ($U \gtrsim W$), which undergoes the transition to metal by application of a perturbation, is used as the top QW layer. Owing to the long distance between the top and bottom QWs, there is no RT effect between the two QWs. The corresponding band diagram and standing waves are illustrated on the right side. The potential well is represented as a black plate, while the existence-probability amplitudes of the metallic QW states are drawn in red for the top QW and in yellow for the bottom QW. Owing to the marginal Mott insulating nature of top QW states, the strongly correlated electrons in the original top QW states become localized, resulting in a localized state (lower Hubbard band) presented in blue. **b** After switching on the RT effect. Owing to the hybridization between the top and bottom QW states, bonding (red curve) and antibonding (yellow curve) states are formed. In this situation, electrons in the top QW will be able to move to the bottom QW via RT. By gaining the spatial degree of freedom, the effective Coulomb repulsion of electrons is weakened and the top QW undergoes the transition from the marginal Mott insulator ($U \gtrsim W$) to the metal ($U < W$).

double QW structures where layers of the strongly correlated conductive oxide SrVO$_3$ (SVO) sandwich a barrier layer of SrTiO$_3$ (STO), a band insulator. Being a highly correlated Fermi-liquid (FL) metal with simple $3d(t_{2g})^1$ configuration[24–30], SVO in ultrathin film grown onto STO is known to form QW states[31,32], and to undergo a thickness-dependent transition from the FL metal to a Mott insulator at a critical film thickness of 2–3 monolayers (ML)[33–35]. A recent theoretical study has predicted that the 2 ML of SVO is at the verge of the Mott insulator, and it can easily become a metal by applying a small perturbation[36].

Thus, as the top marginal Mott-insulating QW, we used a 2-ML SVO layer. As a counterpart, we used a 6-ML SVO for the bottom metallic QW layer, so as to induce the RT effect between two energetically close QW states (Supplementary Note 3). Thereafter, the (2-ML SVO)/($L$-ML STO)/(6-ML SVO) double QW structure is denoted as $V_2T_LV_6$, where $L$ is the thickness of the STO barrier layer. Thus, based on the previously reported structure plot of quantization energies as a function of SVO layer thickness[31–33,37,38], the first quantization level of the top (2-ML SVO) QW states matches the second quantization level of the bottom (6-ML SVO) QW. Furthermore, in the $V_2T_LV_6$ structure, the transition probability of electrons between the top and bottom QWs is also controlled as a function of the STO barrier layer thickness $L$, as schematically presented in Fig. 1.

**RT-driven MIT in double QW structures**. The transition from the marginal Mott-insulating QW states to the metallic QW states induced by the RT effects is visualized by in situ angle-resolved photoemission spectroscopy (ARPES). Figure 2a presents a series of ARPES images of the $V_2T_LV_6$ double QW structures with varying the STO barrier layer thickness ($L = 2, 4, 10$, and $\infty$ ML).

Because these band dispersions have been taken along the Γ–X direction, the ARPES images consist of only the $d_{zx}$ bands of V 3$d$ $t_{2g}$ states in the present experimental geometry (see Supplementary Note 7)[31–33,39,40]. Here, the series of ARPES images are normalized to the incident photon flux; hence, the color scale reflects the change in spectral weight as a function of $L$. Thus, the metallization of top QW will be evidenced by the appearance of a parabolic band at the Γ point near the Fermi level ($E_F$).

As can be seen in the ARPES images of the $V_2T_LV_6$ structure, there are no discernible states near $E_F$, reflecting the Mott-insulating nature of the 2-ML SVO films. The Mott-insulating state of the top 2-ML QW of $V_2T_\infty V_6$ is further confirmed by the appearance of the lower Hubbard band at a binding energy of 1.5 eV (see Fig. 2b and Supplementary Fig. 18)[33,34]. In the $V_2T_4V_6$ double QW, as the STO barrier layer becomes thinner (the transition probability of electrons between the two QW states increases), a faint dispersive feature emerges near $E_F$. Eventually, a metallic band whose dispersion crosses $E_F$ is clearly visible in the $V_2T_2V_6$ double QW structures, demonstrating the metallization of the top 2-ML SVO layer (see also Supplementary Notes 10 and 11). It should be noted that the observed metallic states are at

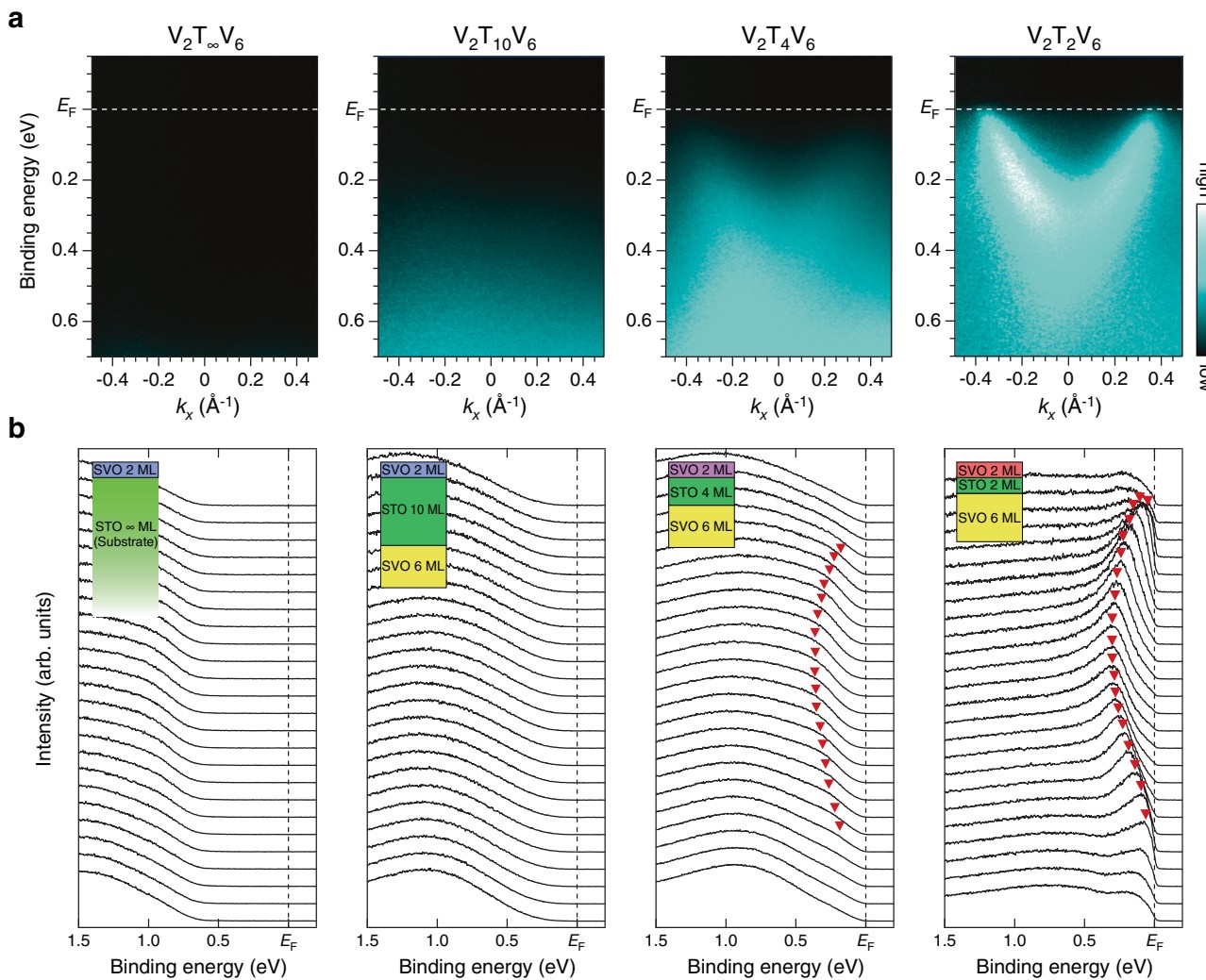

**Fig. 2 Visualization of the metal-insulator transition induced by reduction in the barrier-layer thickness. a** Respective ARPES images for $V_2T_LV_6$ double QW structures with $L$ reducing from $\infty$ to 2. The ARPES data were acquired at a photon energy of $h\nu = 88$ eV along a $k_x$ slice near the Γ point, which includes only the quantized $d_{zx}$ band. Note that the series of ARPES spectra are normalized to the incident photon flux, and the normalized intensity is given by a color scale shown on the right side. **b** Corresponding EDCs to the respective ARPES images. The red filled triangles are guides to the eye for the dispersive features. The broad non-dispersive feature around 1.0 eV is the lower Hubbard band. The inset shows the schematic side views of the corresponding double QW structures.

the top 2-ML SVO layer, since the mean-free-path of the photoelectrons in the present experimental condition is about 0.4–0.6 nm (corresponding to 1.0–1.5 ML in the present case) (refs. [41–43]). Thus, signals from the buried 6-ML SVO layer are negligible in the ARPES results on $V_2T_LV_6$ (Supplementary Note 12).

The emergence of metallic states created by the proximity of two QWs is more clearly seen in the energy distribution curves (EDCs) shown in Fig. 2b, where one observes a systematic evolution of the dispersive metallic states and subsequent reduction in the lower Hubbard band (localized states) in the $V_2T_LV_6$ double QW structures (see also Supplementary Fig. 19). In the top 2-ML SVO layer of the $V_2T_\infty V_6$ and $V_2T_{10}V_6$ double QW structures, no metallic-band-like features are visible near $E_F$ and only non-dispersive features are observed below a binding energy of 1 eV. The existence of the non-dispersive structure is caused by spectral weight transfer from the coherent bands near $E_F$ to the lower Hubbard bands[33,34,44], indicating the localized nature of V 3d electrons in the top 2-ML SVO layer (see Supplementary Note 9). When the barrier layer thickness decreases, a band-like feature appears near $E_F$ for $V_2T_4V_6$, while a weak non-dispersive structure is still visible around 0.8 eV. Eventually, for the thinnest barrier, the band-like feature evolves to a clear dispersive feature crossing $E_F$ (see also Supplementary Note 8).

The spectral behavior observed in the top QW layer of $V_2T_LV_6$ is similar to that previously observed in thickness-dependent Mott transition in SVO QWs[33], suggesting the existence of a Mott transition from localized states to standard QW subbands in 2-ML SVO with decreasing $L$. Thus, the next crucial issue is whether the MIT originates from the RT effects or not. The condition for the RT to occur is that there must be an energetic match between quantized states at both the top and bottom QWs[1]. As seen in the 2-ML QW shown in Fig. 2, the bottom of the conduction band (subband bottom energy), corresponding to its first quantization energy, is estimated to be 320 meV. This value is consistent with the one extrapolated from the structure plot of SVO QW states as a function of SVO layer thickness (see Supplementary Note 3)[31–33,37,38]. To confirm the existence of energetically close QW states for SVO layers at the opposite side of the structure, we investigate the "flip" double QW structure of $V_6T_2V_2$ as shown in Fig. 3a, where we only probe the subband structures of the 6-ML QW due to the photoelectron attenuation[41–43]. As expected from previous works[31–33], the ARPES images consist of the quantized $d_{zx}$ bands with quantum numbers $n = 1$ (band bottom at 500 meV) and $n = 2$ (band bottom at 290 meV) [see Supplementary Notes 3 and 13]. The quantization energy of $n = 2$ states for the 6-ML SVO layer is close to that of the metallic band observed for the 2-ML SVO layer (320 meV). The existence of the energetically close QW states in both the top and bottom QW structures suggests the hybridized nature of the envelope wave functions of the two subbands[45,46], leading to the RT effect between the two QWs.

**Theoretical analysis based on DFT calculations.** The occurrence of the RT effects between the two QW states is further supported by density functional theory (DFT) calculation (Supplementary Note 15). Figure 3 compares the DFT results of $V_2T_2V_6$ QW structures with the ARPES results. Owing to the hybridization between the top and bottom QW states, the DFT calculation shows the formation of four $d_{zx}$-derived subbands ($n' = 1$–4) from the bottom. As a result of the hybridization of the original $n = 1$ of the 2-ML QW and $n = 2$ in the 6-ML QW, these two

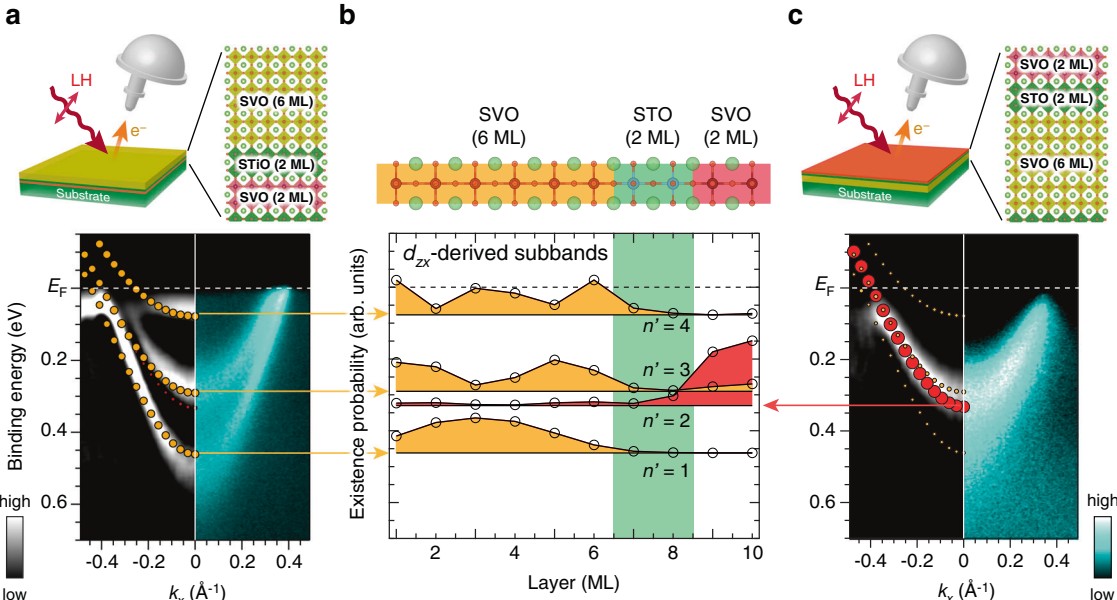

**Fig. 3 Subband structures formed at the $V_2T_2V_6$ heterostructure. a, c** Comparison of ARPES images taken near the Γ point with the DFT calculations for $V_6T_2V_2$ (**a**) and $V_2T_2V_6$ (**c**) double QWs. Raw ARPES intensity plots are shown on the right half side and peak-enhanced plots (see Supplementary Note 8) on the left-half side, together with the DFT calculation. As shown in the schematic layout on the top, electrons near the surface (~0.4–0.6 nm) are detected in the ARPES measurements. Since only $d_{zx}$-derived subbands are detected in this experimental configuration (see Supplementary Note 7), the corresponding subband with predominant $d_{zx}$ character (filled circles) is presented in the calculation for comparison (the subbands for all $t_{2g}$ states are shown in Supplementary Fig. 26). The probability of the electron being detected by ARPES is represented by the size of the filled circles. **b** Existence probabilities for respective $d_{zx}$-derived subbands, which are calculated by Mulliken population analysis, are plotted along the z direction for the $V_2T_2V_6$ heterostructure. The baselines of the existence probabilities (solid black lines) correspond to the quantization energies (subband minimum energies). The existence probability of bonding $n' = 2$ states (red hatched) has central weight at the 2-ML QW, while that of antibonding $n' = 3$ states (yellow hatched) at the 6-ML QW, representing their original character.

energetically close quantization levels form a bonding ($n' = 2$) and an antibonding ($n' = 3$) state in the double QW. From the DFT calculation, the energy difference between quantization levels $n' = 2$ and $n' = 3$ is estimated to be 40 meV. In contrast, the original $n = 1$ and 3 states in the bottom 6-ML QW are not hybridized to any levels in the top QW and remain unchanged. Judging from the existence probability of electrons belonging to each subband as shown in Fig. 3b, the $n' = 2$ subband, which mainly exists in the 2-ML QW, emerges from the original $n = 1$ subband of the 2-ML SVO layer, whereas the other subbands ($n' = 1, 3, 4$) from the original 6-ML SVO QW states. By considering the escape depth of the present ARPES measurements[41–43] and the distribution of existence probabilities, only the $n' = 2$ subband is predominantly detectable in ARPES for the $V_2T_2V_6$ heterostructure, whereas the $n' = 1, 3, 4$ subbands are detectable for the "flip" $V_6T_2V_2$ heterostructure. These features are well reproduced in the ARPES results of Fig. 3a, c, suggesting the occurrence of the RT effects between the top ($n = 1$) and bottom ($n = 2$) QW states.

## Discussion

The DFT calculation provides an important indication regarding the metallization of the top QW states: owing to the close proximity of the two quantization energy levels, only $n' = 2$ and 3 states hybridize with each other. Also, the $n' = 2$ ($n' = 3$) shows weak but finite existence probabilities in the 6-ML SVO layer (2-ML SVO layer) side, and their respective probability maxima spatially overlap. This spread of the existence probability over both sides of the double QW structure results from the hybridization of the corresponding envelope wave functions. Actually, the DFT result demonstrates that about 10% of the $n' = 2$ states spread over the 6-ML SVO QW side in the case of $V_2T_2V_6$ double QW structures. As schematically illustrated in Fig. 1, in this situation, strongly correlated electrons in the top "marginal" Mott QW states of the 2-ML SVO layer ($U \gtrsim W$) achieve additional spatial freedom, since they can move to the bottom metallic 6-ML QW through the RT effects. The RT effects cause the reduction in effective Coulomb potential ($U < W$) in the 2-ML QW. As a result, the marginal Mott insulator undergoes an insulator-to-metal transition.

Although $U/W$ is the most important parameter determining the MIT, it should be bear in mind that the MIT is not determined solely by the value of $U/W$. In particular, in the case of a multi-orbital system, various factors such as electron number, the strength of Hund's coupling, and detailed density-of-states structures determine the boundary of the MIT[36]. Thus, more realistic calculations incorporating such effects, as well as complex interactions of strongly correlated electrons in the QW, are expected. Such calculations are certainly necessary for a quantitative understanding of the observed RT-driven MIT, but this issue remains to be resolved.

The present study demonstrates that the MIT can be controlled by the RT effect in double QW structures of strongly correlated oxides. Our observations offer valuable insight into the quest for novel quantum phenomena using oxide heterostructures[6–11,47–50], since the $U/W$ ratio can be controlled by designing the wave function of their strongly correlated electrons. In addition, from an applied perspective, the MIT control based on the double QW structure studied here has fundamental advantages over conventional FET control[4–8]: the Mott transition may be operated by aligning two quantization levels through the application of a small voltage, and the entire QW will undergo an MIT irrespective of the limitation imposed by Thomas–Fermi screening, as illustrated in Fig. 4. The present demonstration opens an avenue for creating a Mott-transistor operation based

on the quantum RT effects between designed wave functions of strongly correlated electrons.

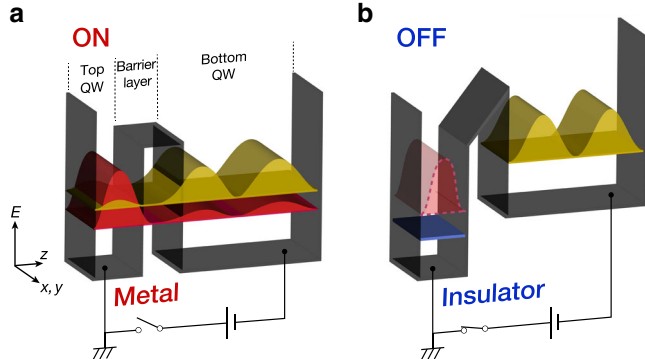

**Fig. 4 Schematic illustration of the Mott transistor based on the metal–insulator transition induced by resonant tunneling effects. a** On states (same as the metallic states shown in Fig. 1b) and **b** off states. Owing to the energy separation of QW states between the top and bottom QWs, there is no RT effect between the two QWs, and hence the top QW becomes a marginal Mott insulator in off states. The Mott transition (on/off operation) is controlled by applying a voltage between the two QWs.

## Methods

**Laser molecular-beam epitaxy**. QW structures were grown on atomically flat (001) surfaces of TiO$_2$-terminated Nb-doped STO substrates in a laser molecular-beam epitaxy chamber connected to an ARPES system at BL-2A of Photon Factory (PF)[30–34] (Supplementary Note 6). During the growth, the thickness was precisely and digitally controlled on the atomic scale by monitoring the intensity oscillation of reflection high-energy electron diffraction spots. The details of the growth conditions of SVO and STO layers[31–34,51] are described in Supplementary Note 4. Note that all double QW structures were fabricated under the same conditions as those of the previously reported SVO/STO heterostructures[31–34,51], wherein atomically flat surfaces and chemically abrupt interfaces formed. The details of the characterizations for the QW structures are given in Supplementary Information. Noted that we carefully characterized the thickness of SVO and STO layers, as well as the chemical abruptness of the SVO/STO interfaces, by analyzing the relative intensities of the relevant core levels just before in situ ARPES measurements (see Supplementary Note 5).

**In situ angle-resolved photoemission spectroscopy**. After growth, the samples were transferred to the ARPES chamber under an ultrahigh vacuum of $10^{-10}$ Torr to avoid the degradation of the sample surfaces on exposure to air (Supplementary Note 6). The ARPES experiments were conducted in situ at a temperature of 20 K using linear horizontal (LH) polarization of the incident light. The incident photon energy was 88 eV. The energy and angular resolutions were respectively set to about 30 meV and 0.3°. The $E_F$ of the samples was calibrated by measuring a gold foil that was electrically connected to the samples. The details of the ARPES measurement setups are given in Supplementary Note 7.

**Electronic structure calculations**. First-principles calculations based on DFT were carried out in the framework of the Perdew–Burke–Ernzerhof-type generalized-gradient approximation[52] using the QMAS code[53] based on the projector augmented-wave method[54] and a plane-wave basis set. The plane-wave cutoff energy was set to 20 Ha. The corresponding Brillouin zone was sampled by $8 \times 8 \times 2$ k-mesh for the self-consistent field calculation. To obtain the electronic density of states, calculations with fixed charges were made at additional **k** points. We have adopted a repeated slab geometry with a vacuum layer (thickness of 11.93247 Å) in between neighboring slabs of the $V_2T_2V_6$ heterostructure as illustrated in the top panel of Fig. 3b. Note that to simplify the calculations, DFT calculations were performed without structural relaxation, setting the cubic lattice constant of both SVO and STO to 3.86494 Å, which is the lattice constant of an SVO crystal. For the comparison with the ARPES results, the subband dispersion ($E_{sub}$) was obtained by multiplying the calculated band structures ($E_{DFT}$) by the band renormalization factor (Z): $E_{sub}(k_{\parallel}) = Z \cdot E_{DFT}(k_{\parallel})$, where $k_{\parallel}$ is the momentum parallel to the surface. The value of Z is inversely proportional to the mass enhancement factor, and the mass enhancement in the subbands is known to be associated with strong interaction among V 3d electrons[24,25,30–33,55–57]. The best fit to the ARPES results gives $Z = 0.55$ (mass enhancement factor ~1.8) for the heterostructure, in good agreement with previous results measured on SVO thin films[31].

## Data availability

All data supporting the key findings of this study are available within the article and its Supplementary Information. All raw data generated during the current study are available from the corresponding author on reasonable request.

## Code availability

All the codes to analyze the experimental data in this study are available from the corresponding author upon reasonable request. The code used for DFT calculation is described in detail in the "Methods". The custom code for DFT analysis is available from S.I. (shoji.ishibashi@aist.go.jp) upon reasonable request.

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

## Acknowledgements

The authors are very grateful to Y. Kuramoto, Y. Matsumoto, and A. Fujimori for their useful discussions and acknowledge A. Wada, N. Hasegawa, D. K. Nguyen, X. Cheng, and E. Sakai for their support in the experiment at PF. This work was financially supported by a Grant-in-Aid for Scientific Research (Nos. 16H02115, 16KK0107, 19H01830, and 20KK0117) from the Japan Society for the Promotion of Science (JSPS), CREST (JPMJCR18T1) from the Japan Science and Technology Agency (JST), and the MEXT Element Strategy Initiative to Form Core Research Center (JPMXP0112101001). A.F.S.-S.

is supported by public grants from the Centre National de la Recherche Scientifique (CNRS), International Research Project (IRP) EXCELSIOR, and the French National Research Agency (ANR), project Fermi-NESt No. ANR-16-CE92-0018. The work performed at KEK-PF was approved by the Program Advisory Committee (proposals 2018S2-004) at the Institute of Materials Structure Science, KEK.

## Author contributions

The samples were grown and characterized by R.Y., T.K., D.S., M. Kobayashi and M.M. ARPES measurements were performed by R.Y., M. Kobayashi, T.K., D.S., M.M., M. Kitamura and K.H. R.Y. and M. Kobayashi analyzed the ARPES data. R.Y., M. Kobayashi, K.Y., A.F.S.-S. and H.K. contributed to the interpretation. S.I. performed the DFT calculation. R.Y., A.F.S.-S. and H.K. wrote the paper with input and discussion from all co-authors. H.K. devised the project and was responsible for its overall planning and direction. All authors discussed the results and commented on the manuscript.

## Competing interests

The authors declare no competing interests.
