## [Peer Review File · Nature Communications]

REVIEWER COMMENTS

Reviewer #1 (Remarks to the Author):

Yukawa and co-authors studied the metal-insulator transition in the quantum well structure of the transition metal oxide heterostructure made through the state-of-the-art laser MBE technology. They made a structure in which the SrVO₃ layer sandwiched the SrTiO₃ layer up and down, and observed through APRES measurement that the metal-insulator transition occurred as the thickness of the SrTiO₃ layer was changed. And they interpreted that the origin of this metal-insulator transition was due to resonant tunneling between SrVO₃ layers through the SrTiO₃ layer, and provided first-principles calculations to support this scenario. The study on the metal-insulator transition in the thin film structure of the transition metal oxide has been an important and interesting research topic in the strongly correlated system community for a long time, and many researchers are still paying attention. Their idea is very interesting, and the quality of their experimental data also satisfies the high standards required by Nature comm. However, this paper has some weaknesses/questions as follows, and if this is resolved, it is well qualified to be published on the Nature comm.

1. In the introduction, the discussion on the Mott-Hubbard metal-insulator transition is too naive. It cannot be denied that U/W is the most important parameter determining the metal-insulator transition, but strictly speaking, the metal-insulator transition is not determined solely by the value of U/W . In particular, in the case of a multi-orbital system, various factors such as electron number, the strength of Hund's coupling, and detailed density-of-states structure determine the metal-insulator transition boundary.
2. They showed ARPES data for several kinds of samples, but they lack support for the quality of the thin films. Through TEM measurement, it is recommended to check whether the thickness of the thin film is well controlled.
3. The authors' DFT calculations provide useful information about the system's resonant tunneling. However, it seems that more calculations are needed for the situation where the tunneling barrier, that is, the SrVO₃ layer is thick. Only then will it be possible to properly show how the resonant tunneling changes when the barrier is adjusted, and the metal-insulator transition is possible accordingly. Also, an important factor of the metal-insulator transition of this material (2ML SrVO₃) is the breakdown of orbital degeneracy. Therefore, what becomes the surface termination layer can be a very important factor. It would be nice to have additional information about this termination dependence through DFT calculation.
4. There is too little information about the details of the DFT calculations. How to get the structure used in the final calculation (how to optimize the force) and what the lattice constant is used are essential information.

Reviewer #2 (Remarks to the Author):

In the present manuscript, Yukawa and coworkers present an ARPES study of a tunneling-based metal-insulator transition (MIT). Taking their prior work to the next level, they show that the correlated metal SVO, which is insulating when grown ultrathin, can undergo an MIT when a barrier with a thicker SVO layer is made thin enough. They rationalize this by the spillage of the wavefunction via (resonant) tunneling, which reduces U via the additional spatial degree of freedom.

Controlling MITs is of significant interest both for fundamental and technological reasons. I am not familiar with a prior demonstration of the concept shown in this manuscript. As such, I find the results original, interesting and significant for a broad audience. The paper is very well written, and other than some minor suggestions I find it suitable to readers from outside of the immediate field. The combination of in-vacuum deposition with ARPES provides a powerful probe into the interesting physics, which is nicely supplemented by first principle calculations. I would add that this work could hugely benefit from electronic transport measurements (e.g. by depositing some protective layer), but the current methodology is sufficient to support the claims here.

As agreed with the editor, I will not comment on the ARPES data since it is beyond my expertise. Therefore, in this department, I have one major comment (the first) and the rest are very minor and technical. If the first comment could be resolved then I believe that this work would be a good fit for the journal and a valuable contribution to the field.

Sincerely,

Lior Kornblum

Technion, Israel

1. Structure. No oxide interface is fully abrupt, and there is always some interface intermixing, which is at the very least on the order of 1 unit cell (uc) on each side of the interface. This is all the more so for energetic deposition such as PLD. As such, especially with the ultrathin barriers, the picture of an abrupt barrier might be naive. V substitution by Ti was shown to induce a filling-controlled MIT,¹ and it can at the very least change the electronic properties of SVO. This possibility should be seriously addressed.
2. Resonant tunneling. In the ‘classic’ definition (from the realm of semiconductors, including Ref. 1) RT involves a double barrier, whereas in the present case there is a single (STO) barrier. While similarities can be drawn between the present case and RT as defined for semiconductor devices, there are still some differences. I ask the authors to clarify and justify the use of ‘resonant tunneling’, versus the more common definition, or perhaps they would like to resort to ‘tunneling’. This shouldn’t take more than a couple of sentences to clarify.
3. Framing. The authors claim that electric field-effect control over the MIT has been unsuccessful. I have two comments on that.

¹ M. Gu, S. A. Wolf, and J. Lu, “Metal-insulator transition in SrTi_{1-x}VxO₃ thin films,” *Appl. Phys. Lett.*, vol. 103, no. 22, p. 223110, 2013.

- A. Field effect control over the MIT and other correlated devices has been successfully achieved in many scenarios using electrolyte gating², and for specific cases via back gating³. Nonetheless, these important achievements have no pretense to result in a practical transistor. The statement should be scaled accordingly (by the way, I would consider citing Zhou⁴ when mentioning MottFET operation).
- B. The report of tunneling as a way to tune the MIT is indeed interesting, but at least in its current presentation it seems to be *static*, i.e. not something that can be switched (similar to strain/chemical-induced bandwidth control, and doping/defect-induced filling control). However in the introduction this is framed as a competing approach to field effect for practical realizations. Please clarify.
4. The concept of the work is very clearly and pedagogically explained in the third paragraph. The emergence of metallicity is explained by the U/W ratio, and the spillage of the wavefunction reduces U. Please comment briefly why is W assumed to be unchanged in this picture; I would expect that the ‘more-3D’ character of the electrons in the thin barrier case would also increase W. One wonders which effect (U vs. W) is more significant in U/W.
5. Related literature. Especially since the present work is focused on spectroscopic analysis, I’m missing complementary literature in the introduction and discussion, mostly dealing with transport analysis of SVO. The pioneering works⁵ of the PSU group should be acknowledged, perhaps at the mention of the Fermi liquid character of the material. Moreover, the thickness induced MIT, dubbed as ‘dimensionality crossover’⁶ by the U. Virginia group should be mentioned when discussing thickness related MIT (note that their thickness threshold is an order of magnitude higher, probably due to the contribution of defects in the localization of the electrons).
6. Vacuum and terminology. While this might be petty, the authors should be consistent and accurate in the language of ‘in situ’ versus ‘in vacuum’ transfer. The supplement is suggesting that transfer occurs within the same UHV system; if this is done by a suitcase please write so explicitly. Also, the authors mention in the supplement collecting core spectra to ensure the surface cleanliness. It would be quite nice to include them in the supplement as well. It would further be useful for the community to observe the core levels of the material without ambient exposure, as a clean reference.
7. Growth details. While it is not unthinkable to send the readers to older publications, given that there’s no space limit on the supplement, please provide 1-2 sentences with basic parameters such as the substrate temperature, PO₂, targets, frequency etc.

² C. Leighton, “Electrolyte-based ionic control of functional oxides,” *Nat. Mater.*, vol. 18, no. 1, pp. 13–18, 2019.

³ A. D. Caviglia, M. Gabay, S. Gariglio, N. Reyren, C. Cancellieri, and J.-M. Triscone, “Tunable Rashba Spin-Orbit Interaction at Oxide Interfaces,” *Phys. Rev. Lett.*, vol. 104, no. 12, p. 126803, Mar. 2010.

⁴ Y. Zhou and S. Ramanathan, “Correlated Electron Materials and Field Effect Transistors for Logic: A Review,” *Crit. Rev. Solid State Mater. Sci.*, vol. 38, no. 4, pp. 286–317, 2013.

⁵ E.g. J. A. Moyer, C. Eaton, and R. Engel-Herbert, “Highly Conductive SrVO₃ as a Bottom Electrode for Functional Perovskite Oxides,” *Adv. Mater.*, vol. 25, no. 26, pp. 3578–3582, 2013

L. Zhang *et al.*, “Correlated metals as transparent conductors,” *Nat Mater*, vol. 15, p. 204, 2015 and others.

⁶ M. Gu, S. A. Wolf, and J. Lu, “Two-Dimensional Mott Insulators in SrVO₃ Ultrathin Films,” *Adv. Mater. Interfaces*, vol. 1, no. 7, pp. 1300126, 2014.

Reviewer #3 (Remarks to the Author):

The manuscript of Yukawa et al describes the occurrence of a metal-insulator transition in double quantum wells (QW) of the SVO-STO-SVO structure. Through detailed ARPES measurements and calculations, they clarify that this is due to the presence/absence of the resonant tunneling between the electronic states between the 2 QW. I think the work has been done nicely and the analysis seems to be adequate. However, I suggest the authors to make the changes below. In my opinion, the present manuscript does not seem to be so novel and does not deserve publication in Nat. Comm. at the present stage. The reasons are also written below.

1. The concept that the wave function connects through thin tunneling barriers itself is not novel at all. What the authors claim to be new is that this is seen in QW of oxide materials. But there is no surprise in this. Also, the authors have only shown this resonant tunneling for one particular combination of the QW thickness V2TLV6. The authors should show at least one different thickness configuration of the QW for the same SVO/STO/SVO system or use different materials and observe the same phenomena to say that this can be universally seen in oxide materials.

2. Related to the above comment, why does the STO thickness need to be 2 ML, not 1 or 3? Intuitively, I would expect that 1 ML STO as the interval layer should also work. What determines the length scale of the insulating layer needed for the resonant tunneling?

3. The authors have only showed the ARPES data taken at $h\nu=88\text{eV}$. Since they are doing the experiments at synchrotron radiation, why not show the data taken at other photon energies? They claim that "It should be noted that the observed metallic states are at the top 2-ML SVO layer, since the mean-free-path of the photoelectrons in the present experimental condition is about 0.4–0.6 nm (corresponding to 1.0–1.5 ML in the present case)(refs 24–26)." in lines 114-116. But shouldn't they seem some difference by using the photon energy of 30-2000eV that is available in this beamline?

4. The authors compare the present results with the field-effect transistors of semiconductors. I would like to actually see the transport data of the present systems (the difference of the conductivity between metallic state V2T2V6 and marginal Mott state V2T10V6).

5. In Fig. 3, the authors compare the data of vacuum / SVO 6ML / STO 2ML/ SVO 2ML /STO substrate and vacuum / SVO 2ML / STO 2ML/ SVO 6ML /STO substrate (although the vacuum and STO substrate are not explicitly written). Are these two comparable? Since the substrate and vacuum should not have the same contribution in the confinement of the QW, I wonder if this comparison is really reasonable or not. Please make some comments on this point.

In summary, I do not see significant novel findings in the present manuscript for publication in Nat. Comm. at this stage. After the authors address the above comments properly and make modifications, I would be happy to review it again.

Reply to Reviewer #1

Yukawa and co-authors studied the metal-insulator transition in the quantum well structure of the transition metal oxide heterostructure made through the state-of-the-art laser MBE technology. They made a structure in which the SrVO₃ layer sandwiched the SrTiO₃ layer up and down, and observed through APRES measurement that the metal-insulator transition occurred as the thickness of the SrTiO₃ layer was changed. And they interpreted that the origin of this metal-insulator transition was due to resonant tunneling between SrVO₃ layers through the SrTiO₃ layer, and provided first-principles calculations to support this scenario. The study on the metal-insulator transition in the thin film structure of the transition metal oxide has been an important and interesting research topic in the strongly correlated system community for a long time, and many researchers are still paying attention. Their idea is very interesting, and the quality of their experimental data also satisfies the high standards required by Nature comm. However, this paper has some weaknesses/questions as follows, and if this is resolved, it is well qualified to be published on the Nature comm.

Reply: We are grateful to Reviewer #1 for the strong recommendation for publication of this manuscript in Nat. Commun., as well as for the high level of appreciation expressed for the present work. We greatly appreciate him/her not only for his/her excellent insight into the importance of our study with regard to the strongly correlated electron systems, but also for his/her deep understanding of current topics in this field. In accordance with the reviewer's suggestions, we have revised the manuscript as detailed below (the respective changes in the manuscript are highlighted in red).

1. In the introduction, the discussion on the Mott-Hubbard metal-insulator transition is too naive. It cannot be denied that U/W is the most important parameter determining the metal-insulator transition, but strictly speaking, the metal-insulator transition is not determined solely by the value of U/W . In particular, in the case of a multi-orbital system, various factors such as electron number, the strength of Hund's coupling, and detailed density-of-states structure determine the metal-insulator transition boundary.

Reply: We agree with Reviewer #1's comments. We have added the following additional discussion in the main text to remind readers of those comments:

Although U/W is the most important parameter determining the MIT, it should be bear in mind that the MIT is not determined solely by the value of U/W . In particular, in the case of a multi-orbital system, various factors such as electron number, the strength of Hund's coupling, and detailed density-of-states structure determine the boundary of the MIT³⁶. Thus, more realistic calculations incorporating such effects, as well as complex interactions of strongly correlated electrons in the QW,

are expected. Such calculations are certainly necessary for a quantitative understanding of the observed RT-driven MIT, but this issue remains to be resolved.

2. They showed ARPES data for several kinds of samples, but they lack support for the quality of the thin films. Through TEM measurement, it is recommended to check whether the thickness of the thin film is well controlled.

Reply: As the reviewer pointed out, it is crucial to determine whether the thickness of the thin film is well controlled. This point was taken very seriously in this study. We present evidence that each layer in the prepared double QW structure is digitally controlled. Figure R1 (Fig. S4 in Supplementary Note 5) shows the Ti 2*p* core-level spectra of SrVO₃/SrTiO₃ with varying SrVO₃ overlayer thickness *t* as well as a SrTiO₃ substrate as a reference. The plot of the Ti 2*p* core-level intensity *I*_{Ti} as a function of *t* in Fig. S4b exhibits excellent agreement with the photoemission attenuation function, suggesting the successful digital control of the SrVO₃ layer thickness. Furthermore, from the detailed simulation, the interdiffusion length of the constituent ions is estimated to be less than 0.2 nm. These results demonstrate that the SrVO₃ layer is precisely controlled at the atomic scale.

For the SrTiO₃ barrier layer, we also evaluated the thickness of a barrier SrTiO₃ layer sandwiched between SrVO₃ layers by core-level spectra. Figure R2 (Fig. S6) shows the plot of *I*_{Ti} of V₂T_LV₆ as a function of *L*, and it compares the results with a simulation based on the photoelectron attenuation function under the assumption of a chemically abrupt interface. The excellent agreement between the experimental and calculated results indicates the successful digital control of the SrTiO₃ barrier layer in the studied double QW structures.

Using the core-level analysis, we demonstrated that the thickness of the thin film was well controlled at the atomic scale. It should be noted that just before ARPES measurements, we carefully characterized the thickness of the SrVO₃ and SrTiO₃ layers by analyzing the relative intensities of the relevant core levels. After confirming that the constituent layer thicknesses of the double QW structures were precisely controlled at the atomic level, we performed *in situ* ARPES measurements. We have added these core-level analyses to the Supplementary Information, as well as the detailed explanations regarding the atomically and chemically abrupt SrVO₃/SrTiO₃ interfaces (Supplementary Note 5. Formation of a chemically abrupt interface; 5.1 Core-level analysis for SrVO₃/SrTiO₃ interface and 5.2 Core-level analysis for digital control of SrTiO₃ layers).

FIG. R1: **a**, Thickness dependence of Ti $2p_{3/2}$ core-level spectra measured at $h\nu = 800$ eV for SrVO₃/Nb:SrTiO₃(001) QW structures. Each spectrum is normalized to the incident photon flux; hence, the intensity reduction with increasing SrVO₃ overlayer thickness t reflects the attenuation of the Ti $2p$ signal from the buried SrTiO₃ by the overlayer. **b**, Plot of relative intensities of the background-subtracted Ti $2p_{3/2}$ core-level spectra I_{Ti} as a function of t . Note that the excellent agreement between the experiment and calculation also indicates the successful digital control of overlayer thickness. The inset shows the logarithm plot of I_{Ti} with respect to t in comparison with the simulation curves that assume an intermixing of V and Ti ions at the interface with interdiffusion lengths d of 0.2–0.7 nm. From the comparison, it is clear that the interdiffusion length of the present film is less than 0.2 nm, which is approximately half of the c -axis length of ~ 0.39 nm.

FIG. R2: Plots of the intensity of Ti $2p$ core levels as a function of L (open circles with error bars), together with the simulated curve derived from the photoelectron attenuation function (a black line).

According to the constructive suggestions from the reviewer, we also performed TEM observations of the double QW structures. We have added the image and detailed explanation in the Supplementary Information (Supplementary Note 5. Formation of a chemically abrupt interface; 5.3 Characterization of double QW structures by HAADF-STEM measurements). Figure S7 shows the high-angle annular dark-field scanning TEM (HAADF-STEM) image of the amorphous SrTiO₃/ SrVO₃ (20 ML)/ SrTiO₃ (2 ML)/ SrVO₃ (6 ML)/ Nb:SrTiO₃ substrate structure. Although the TEM measurements confirmed the coherent growth of SrVO₃ and SrTiO₃ layers on Nb:SrTiO₃ substrates without the formation of any dislocations, it is hard to distinguish between SrVO₃ and SrTiO₃ layers. This is due to the following reasons: 1) Since HAADF-STEM measures the difference in atomic weight Z, there is no detectable difference between Ti (Z = 22) and V (Z = 23) ions; and 2) both oxides share a common A-site composition (SrO). Therefore, we employed the core-level analysis for the thickness control of each oxide layer.

3. The authors' DFT calculations provide useful information about the system's resonant tunneling. However, it seems that more calculations are needed for the situation where the tunneling barrier, that is, the SrVO₃ layer is thick. Only then will it be possible to properly show how the resonant tunneling changes when the barrier is adjusted, and the metal-insulator transition is possible accordingly.

Reply: According to the constructive comments from the reviewer, we performed additional DFT calculations for V₂T_LV₆ (L = 1, 2, 3, 4, and 10). We have added the DFT results in the Supplementary Information, together with detailed explanations (Supplementary Note 15. DFT calculations for subband structure formed in the V₂T_LV₆ heterostructures; 15.1 Barrier-layer-thickness dependence). In the DFT results, we confirmed the systematic evolution of resonant tunneling effects between the two energetically close QW states (i.e., n' = 2 and 3 in Fig. 3) as a function of the SrTiO₃ barrier layer thickness L. In contrast, the other QW states that do not contribute to resonant tunneling are not hybridized to any levels and remain unchanged. We believe that these additional DFT calculations further strengthen the validity of our conclusions.

Also, an important factor of the metal-insulator transition of this material (2ML SrVO₃) is the breakdown of orbital degeneracy. Therefore, what becomes the surface termination layer can be a very important factor. It would be nice to have additional information about this termination dependence through DFT calculation.

Reply: Reviewer #1 suggests that we demonstrate the termination dependence of electronic structures. Following this constructive suggestion, we performed DFT calculations for different surface terminations. Figures S27 and S28 show the DFT calculations for the a) VO₂-terminated (the same as those in Fig. 3a) and b) SrO-terminated V₂T₂V₆. At first glance, there is no significant difference between the two. However, a closer

look reveals that the QW states in SrO-terminated 2-ML SVO QW ($n_{2\text{ML}}$) rigidly shift toward lower binding energies. The upward shift of $n_{2\text{ML}} = 1$ for SrO-terminated $\text{V}_2\text{T}_2\text{V}_6$ may arise from the charge transfer from the top 2-ML QW to the topmost SrO layer and the resultant reduction of the population (chemical potential shift) in the original 2-ML QW structures. Consequently, the resonant tunneling (RT) effect occurs between $n_{2\text{ML}} = 1$ and $n_{6\text{ML}} = 3$ ($n_{2\text{ML}} = 2$ and $n_{6\text{ML}} = 5$) owing to the proximity of the two quantization energy levels, as shown in Fig. S27, although the RT effect in SrO termination is significantly weaker than that in VO_2 termination, reflecting the larger difference in the quantization energies of the original QW states.

It should be noted that the termination layer of the double QW structure is a VO_2 atomic layer, since we used TiO_2 -terminated SrTiO_3 substrates [Y. Okada *et al. Phys. Rev. Lett.* **119**, 086801 (2017)]. Thus, we have compared the ARPES results with the VO_2 -terminated $\text{V}_2\text{T}_2\text{V}_6$ structure. Indeed, as can be seen in Fig. 3, the experimental data agree much better with the VO_2 -terminated $\text{V}_2\text{T}_2\text{V}_6$ structure.

In order to provide additional information regarding the termination dependence, we have added the additional DFT calculations to the Supplementary Information (Supplementary Note 15. DFT calculations for subband structure formed in the \$\text{V}_2\text{T}_2\text{V}_6\$ heterostructures; 15.2 Termination-layer dependence).

4. There is too little information about the details of the DFT calculations. How to get the structure used in the final calculation (how to optimize the force) and what the lattice constant is used are essential information.

Reply: We have added a detailed explanation about the DFT calculations in the Methods section (Electronic structure calculations) as follows. In the present study, we used a fixed lattice parameter to focus upon the resonant tunneling phenomena. In the near future, we will perform more detailed calculations by optimizing the force.

Electronic structure calculations. First-principles calculations based on DFT were carried out in the framework of the Perdew-Burke-Ernzerhof-type generalized-gradient approximation⁵² using the QMAS code⁵³ based on the projector augmented-wave method⁵⁴ and a plane-wave basis set. The plane-wave cutoff energy was set to 20 Ha. The corresponding Brillouin zone was sampled by $8 \times 8 \times 2$ k -mesh for the self-consistent field calculation. To obtain the electronic density of states, calculations with fixed charges were made at additional k points. We have adopted a repeated slab geometry with a vacuum layer (thickness of 11.93247 Å) in between neighboring slabs of the $\text{V}_2\text{T}_2\text{V}_6$ heterostructure as illustrated in the top panel of Fig. 3b. Note that to simplify the calculations, DFT calculations were performed without structural relaxation, setting the cubic lattice constant of both SVO and STO to 3.86494 Å, which is the lattice constant of an SVO crystal.

We believe that the additional experimental data and DFT results will convince readers of the solidity of our data analysis and hence the validity of our conclusions.

Reply to Reviewer #2 (Prof. Lior Kornblum):

In the present manuscript, Yukawa and coworkers present an ARPES study of a tunneling-based metal-insulator transition (MIT). Taking their prior work to the next level, they show that the correlated metal SVO, which is insulating when grown ultrathin, can undergo an MIT when a barrier with a thicker SVO layer is made thin enough. They rationalize this by the spillage of the wavefunction via (resonant) tunneling, which reduces U via the additional spatial degree of freedom. Controlling MITs is of significant interest both for fundamental and technological reasons. I am not familiar with a prior demonstration of the concept shown in this manuscript. As such, I find the results original, interesting and significant for a broad audience. The paper is very well written, and other than some minor suggestions I find it suitable to readers from outside of the immediate field. The combination of in-vacuum deposition with ARPES provides a powerful probe into the interesting physics, which is nicely supplemented by first principle calculations. I would add that this work could hugely benefit from electronic transport measurements (e.g. by depositing some protective layer), but the current methodology is sufficient to support the claims here.

As agreed with the editor, I will not comment on the ARPES data since it is beyond my expertise. Therefore, in this department, I have one major comment (the first) and the rest are very minor and technical. If the first comment could be resolved then I believe that this work would be a good fit for the journal and a valuable contribution to the field.

Reply: We are very grateful to Prof. Kornblum for the strong recommendation for publication of this manuscript in Nat. Commun., as well as for the high level of appreciation expressed for the present work. We sincerely thank him not only for his considerable insight into the importance of our study regarding the development of next-generation electronics, but also for his deep understanding of current topics in solid-state physics. In accordance with his valuable suggestions for the improvement of our manuscript, we have revised the manuscript as detailed below (the respective changes in the manuscript are highlighted in red).

1. Structure. No oxide interface is fully abrupt, and there is always some interface intermixing, which is at the very least on the order of 1 unit cell (uc) on each side of the interface. This is all the more so for energetic deposition such as PLD. As such, especially with the ultrathin barriers, the picture of an abrupt barrier might be naive. V substitution by Ti was shown to induce a filling-controlled MIT[1], and it can at the very least change the electronic properties of SVO. This possibility should be seriously addressed.

Reply: Referee #2 is concerned about the influence of mixing at the interface. This point is one that we took most seriously in this study. We have confirmed that the prepared double QW structures feature atomically and

chemically abrupt SrVO₃/SrTiO₃ interfaces with interdiffusion length of less than 0.2 nm, and that the possible intermixing has negligible influence on our observations, by following characterizations:

1. Core-level intensity analysis for SrVO₃/ SrTiO₃ interfaces to confirm the formation of chemically abrupt interfaces.
2. Core-level intensity analysis for V₂T_LV₆ double QW structures to confirm the digital control of the SrTiO₃ barrier layers.
3. *In situ* ARPES characterization for the invariance of QW states between vacuum/ SrVO₃/ SrTiO₃ and SrTiO₃/ SrVO₃/ SrTiO₃.

These results are discussed below.

1. Core-level intensity analysis for SrVO₃/ SrTiO₃ interfaces to confirm the formation of chemically abrupt interfaces: We present evidence that the prepared double QW structures feature chemically abrupt SrVO₃/SrTiO₃ interfaces. Figure R1 (Fig. S4 in Supplementary Note 5) shows the Ti 2*p* core-level spectra of SrVO₃/SrTiO₃ with varying SrVO₃ overlayer thickness *t*, as well as a SrTiO₃ substrate as a reference. The plot of the resultant Ti 2*p* core-level intensity *I_{Ti}* as a function of *t* in Fig. R1 (Fig. S4) exhibits excellent agreement with the photoemission attenuation function. The detailed simulation reveals that the interdiffusion length of the constituent ions is estimated to be less than 0.2 nm (0.5 ML). Since both oxides share a common A-site composition (SrO), these results indicate the formation of a chemically abrupt interface between SrVO₃ and SrTiO₃.

2. Core-level intensity analysis for V₂T_LV₆ double QW structures to confirm the digital control of the SrTiO₃ barrier layers: Figure R2 (Fig. S6 in Supplementary Note 5) show the core-level spectra of V₂T_LV₆ double QW structures with varying SrTiO₃ barrier-layer thickness *L*. The comparison of the core-level intensity with a simulation based on the photoelectron attenuation function assuming a chemically abrupt interface shows an excellent agreement. The excellent agreement between the experimental and calculated results indicates the successful digital control of the SrTiO₃ barrier layer in the present double QW structures, as well as the formation of a chemically abrupt interface within the experimental margins.

3. *In situ* ARPES characterization for the invariance of QW states between vacuum/ SrVO₃/ SrTiO₃ and SrTiO₃/ SrVO₃/ SrTiO₃: Figure S23 (in Supplementary Note 13) shows the comparison of the ARPES images between vacuum/ 6-ML SrVO₃ and 4-ML SrTiO₃/ 6-ML SrVO₃ (T₄V₆) QW structures. We observed very weak but distinct QW states from the bottom QW in T₄V₆, when the dynamic range in the T₄V₆ image is in expanded scale (x20). As can be seen in Fig. S23c, it is evident that there is little change in the quantization levels (*n* = 1 and 2) of both QW structures, demonstrating that the SrTiO₃ layer acts as a vacuum in the present QW structures. The invariance of the QW states at the surface (vacuum/SrVO₃ interface) and the interface (SrTiO₃/SrVO₃)

provides further evidence that the possible intermixing at the SrTiO₃/SrVO₃ interfaces has negligible influence on our observations.

It should be noted that we carefully characterized the thicknesses of SrVO₃ and SrTiO₃ layers and the chemical abruptness of SrVO₃/SrTiO₃ interfaces by analyzing the relative intensities of the relevant core levels just before ARPES measurements. After confirming that the constituent layer thickness of the double QW structures was precisely controlled at the atomic level, we performed *in-situ* ARPES measurements. Furthermore, the quantized energies observed by *in-situ* ARPES are in excellent agreement with the prediction from the phase-shift quantization rule (see Fig. S1); this provides another evidence that possible intermixing does not affect the standing waves in the SVO QW structures. We have emphasized this fact in the main text (lines 229–232).

We have added these data to the Supplementary Information and provided the detailed explanations regarding the formation of atomically and chemically abrupt SrVO₃/SrTiO₃ interfaces (Supplementary Note 5. Formation of a chemically abrupt interface), where possible intermixing does not affect the standing waves in SrVO₃ QW structures (Supplementary Note 13. Quantum confinement at vacuum/SrVO₃ and SrTiO₃/SrVO₃ interfaces).

2. Resonant tunneling. In the 'classic' definition (from the realm of semiconductors, including Ref. 1) RT involves a double barrier, whereas in the present case there is a single (STO) barrier. While similarities can be drawn between the present case and RT as defined for semiconductor devices, there are still some differences. I ask the authors to clarify and justify the use of 'resonant tunneling', versus the more common definition, or perhaps they would like to resort to 'tunneling'. This shouldn't take more than a couple of sentences to clarify.

Reply: We thank the reviewer for the helpful comment regarding the improvement of our manuscript. As the reviewer pointed out, in the conventional resonant tunneling phenomena of compound semiconductors, a QW is present between two tunnel barriers with doped contacts (or metal electrodes) on either side to form electron reservoirs. When a voltage is applied, a current flows only when electrons can tunnel through the subband state in the QW, which induces negative differential resistance. In a double QW structure with triple barriers, the two QW states with discrete energy levels also exhibit resonant tunneling between the two QW structures when the discrete energies match each other [Y. Majima et al., JACS **135**, 14159 (2013)]. Although one of the two barriers is a vacuum for the top QW structures in the present double QW structures, the MIT is induced by the significant spread of the wavefunction owing to the energy matching between the QW states ($n = 2$ in the bottom QW structure and $n = 1$ in the top QW structure). Hence, we use the term resonant tunneling.

In order to avoid unnecessary confusion to readers, we have added the following footnote to the Supplementary Information (Supplementary Note 1. Definition of “resonant tunneling”).

In general, resonant tunneling is a phenomenon that an electron passes through the barrier layers in quantum well (QW) structures without energy decay; it occurs when the energy of an incoming electron matches that of an electron confined in the two potential barriers¹. Although one of the two barriers is a vacuum for the top QW structures in the present double QW structures, the metal-insulator transition occurs through the significant spread of the wavefunction owing to the energy matching between the QW states ($n = 2$ in the bottom QW structure and $n = 1$ in the top QW structure). Hence, we use the term “resonant tunneling”.

3. Framing. The authors claim that electric field-effect control over the MIT has been unsuccessful. I have two comments on that.

A. Field effect control over the MIT and other correlated devices has been successfully achieved in many scenarios using electrolyte gating [2], and for specific cases via back gating [3]. Nonetheless, these important achievements have no pretense to result in a practical transistor. The statement should be scaled accordingly (by the way, I would consider citing Zhou [4] when mentioning MottFET operation).

Reply: We thank the reviewer for his helpful comments for the improvement of our manuscript, as well as for suggesting the appropriate references. According to the reviewer’s suggestions, we have revised the sentence as follows, and we have cited these works together with the related works (Refs. 6, 7, 12–21 in the revised manuscript).

Some strongly correlated electron materials naturally show a metal-to-insulator transition (MIT)^{2,3}. It would be thus highly desirable to control such an MIT in the same way as it is done for semiconductors. Numerous efforts have been made to demonstrate the FET control of the MIT⁴⁻⁸, as well as of rich quantum phase transitions in strongly correlated electron materials⁹⁻¹⁵. For example, field effect control over the MIT has been successfully achieved in many scenarios using electrolyte gating in the electric double-layer transistor^{7,8}, although such gate-controlled MIT is challenging because of the possible electrochemical reaction in the ionic liquid gate^{16,17}. Meanwhile, the FET control of other correlated devices has been also achieved for specific cases via back gating¹⁸⁻²⁰. Nonetheless, these important achievements have no pretense to result in a practical transistor^{6,7,21}. In fact, such approach faces fundamental difficulties. One is the insufficient carrier density that can be induced by the electric field to cause a filling-controlled MIT (Mott transistor operation)^{4,5}. The other is the shortness of the Thomas-Fermi screening length due to $10^{22} - 10^{23} \text{ cm}^{-3}$ mobile carriers,

which limits the conductive area where MIT occurs^{7,8}. Thus, realistic future applications of the Mott transistor call for a different principle of controlling the MIT²².

B. The report of tunneling as a way to tune the MIT is indeed interesting, but at least in its current presentation it seems to be static, i.e. not something that can be switched (similar to strain/chemical-induced bandwidth control, and doping/defect-induced filling control). However in the introduction this is framed as a competing approach to field effect for practical realizations. Please clarify.

Reply: We thank the reviewer for his helpful advice regarding the improvement of our manuscript. According to this advice, we have added a schematic illustration of the concept of the Mott transistor, which is based on the MIT induced by resonant tunneling (Fig. 4 in the main text). We also display this below for convenience.

Figure 4 | Schematic illustration of the Mott transistor based on the metal-insulator transition induced by resonant tunneling effects. **a**, On states (same as the metallic states shown in Fig. 1b) and **b**, Off states. Owing to the energy separation of QW states between the top and bottom QWs, there is no RT effect between the two QWs, and hence the top QW becomes a marginal Mott insulator in off states. The Mott transition (on/off operation) is controlled by applying a voltage between the two quantum wells.

4. The concept of the work is very clearly and pedagogically explained in the third paragraph. The emergence of metallicity is explained by the U/W ratio, and the spillage of the wavefunction reduces U . Please comment briefly why is W assumed to be unchanged in this picture; I would expect that the 'more-3D' character of the electrons in the thin barrier case would also increase W . One wonders which effect (U vs. W) is more significant in U/W .

Reply: According to the Mott-Hubbard theory, W is defined as the kinetic energy of an electron moving between sites, whereas U is defined as the on-site Coulomb repulsion between electrons. In the present double QW structures, W is primarily determined by the spread of the wavefunction in the plane, because the spread area of the wavefunction perpendicular to the plane is much smaller than in the plane owing to the quantum confinements. Therefore, the spillage of the wavefunction along the direction perpendicular to the plane negligibly influences the W value. Meanwhile, the effective U is determined by the Coulomb integral; hence, the spillage of the wavefunction largely affects U .

In order to avoid unnecessary confusion to readers, we have added the following footnote in the Supplemental Information, explaining why W is assumed to be unchanged (Supplementary Note 2. Why \$W\$ is assumed to be unchanged).

According to the Mott-Hubbard theory², the bandwidth (W) is defined as the kinetic energy of an electron moving between sites, whereas the Coulomb interaction (U) is defined as the on-site Coulomb repulsion between electrons. In the QW structures considered here, W is primarily determined by the spread of the wavefunction in the plane, because the spread of the wavefunction perpendicular to the plane is considerably smaller than that in the plane, owing to the quantum confinements. Therefore, the spillage of the wavefunction along the direction perpendicular to the plane has a negligible influence on the W value. Meanwhile, the effective U is determined by the Coulomb integral; hence, the spillage of the wavefunction largely affects U .

5. Related literature. Especially since the present work is focused on spectroscopic analysis, I'm missing complementary literature in the introduction and discussion, mostly dealing with transport analysis of SVO. The pioneering works[5] of the PSU group should be acknowledged, perhaps at the mention of the Fermi liquid character of the material. Moreover, the thickness induced MIT, dubbed as 'dimensionality crossover'[6] by the U. Virginia group should be mentioned when discussing thickness related MIT (note that their thickness threshold is an order of magnitude higher, probably due to the contribution of defects in the localization of the electrons).

Reply: We are very grateful to the reviewer for pointing us to the pieces of appropriate literature. We have cited these works in the main text accordingly, together with the related works (Refs. 26-29 for the Fermi liquid character of SrVO₃ and Ref. 35 for thickness-related MIT in the revised manuscript).

6. Vacuum and terminology. While this might be petty, the authors should be consistent and accurate in the

language of 'in situ' versus 'in vacuum' transfer. The supplement is suggesting that transfer occurs within the same UHV system; if this is done by a suitcase please write so explicitly. Also, the authors mention in the supplement collecting core spectra to ensure the surface cleanliness. It would be quite nice to include them in the supplement as well. It would further be useful for the community to observe the core levels of the material without ambient exposure, as a clean reference.

Reply: We transferred the sample within the same ultrahigh vacuum (UHV) system; that is, the “*in-situ* photoelectron spectrometer – laser molecular beam epitaxy (MBE)” system, in which a photoemission chamber is connected to laser-MBE equipment in an UHV. This system consists of four interconnected chambers: sample entry, sample preparation, laser MBE, and photoemission. The photoemission chamber is connected to the beamline. The four chambers are connected to each other in an UHV and each chamber can be isolated using gate valves.

To demonstrate the cleanliness of the surface transferred within this UHV system (*in-situ* transfer), we have added core-level spectra (survey scan) in the Supplementary Information (Fig. S9). These spectra were measured just before the ARPES measurements. No detectable C 1s signal was observed in the core-level photoemission spectra, guaranteeing the high surface cleanliness required for our spectroscopic measurements.

In order to avoid unnecessary confusion to readers, we have added the detailed explanation of our instruments in the revised supplementary information (Supplementary Note 6. “*In situ* photoelectron spectrometer – laser molecular beam epitaxy” system).

7. Growth details. While it is not unthinkable to send the readers to older publications, given that there's no space limit on the supplement, please provide 1-2 sentences with basic parameters such as the substrate temperature, PO₂, targets, frequency etc.

Reply: According to the reviewer's useful suggestion, we have added the following detailed growth conditions in Supplementary Note 4.

“Sintered SrVO₃ and SrTiO₃ pellets were used as ablation targets. An Nd-doped yttrium aluminum garnet laser was used for target ablation in its frequency-tripled mode ($\lambda = 355$ nm) at a repetition rate of 1 Hz. During the deposition of both layers, the substrate temperature was maintained at 900°C, and the oxygen pressure was maintained at less than 10⁻⁸ Torr.”

Reply to Reviewer #3:

The manuscript of Yukawa et al describes the occurrence of a metal-insulator transition in double quantum wells (QW) of the SVO-STO-SVO structure. Through detailed ARPES measurements and calculations, they clarify that this is due to the presence/absence of the resonant tunneling between the electronic states between the 2 QW. I think the work has been done nicely and the analysis seems to be adequate. However, I suggest the authors to make the changes below. In my opinion, the present manuscript does not seem to be so novel and does not deserve publication in Nat. Comm. at the present stage. The reasons are also written below.

Reply: We thank Reviewer #3 for his/her invaluable comments and for his/her recommendation for publication in Nat. Comm. after revision. Based on the suggestions, we have performed additional ARPES experiments and data analyses, and confirmed the validity of our previous conclusions. These additional data and analyses indisputably demonstrate our most interesting conclusions; namely, the occurrence of MIT via resonant tunneling between the two QWs. We have revised the manuscript according to the useful suggestions, including the results of the additional experiments. Revisions of manuscript are as follows (the respective changes in the manuscript are highlighted in red).

1. The concept that the wave function connects through thin tunneling barriers itself is not novel at all. What the authors claim to be new is that this is seen in QW of oxide materials. But there is no surprise in this.

Reply: As the reviewer commented, the observation that the wavefunction connects through thin tunneling barriers in the oxide double QW is not novel, although it is the first observation in strongly correlated oxides. The most important achievement of this study is the finding that the metal-insulator transition (MIT) can be controlled by resonant tunneling (RT) effects in double QW structures of strongly correlated oxides: the top QW undergoes the transition from the Mott insulator to the metal via the particular wavefunction connections through thin tunneling barriers.

Here, we would like to stress the impact of this successful MIT control by using the RT effects in double QW structures of strongly correlated oxides upon this field. Similar to semiconductor-based field-effect transistors (FETs), which burst the information and technology revolution, controlling the MIT via an external stimulus in strongly correlated materials is a key technological goal for applications in future electronic devices. However, despite intensive efforts worldwide, such a "Mott transistor" has not been realized so far, mainly because the standard control through FET faces severe difficulties when applied to the MIT. Hence, a radically different approach is needed.

In this study, we demonstrate a new approach to control the MIT using the RT effects in the double QW structures of strongly correlated oxides. The MIT control based on the double QW structure studied here has fundamental advantages over conventional FET control. Thus, the present demonstration opens an avenue for achieving Mott-transistor operations based on the quantum RT effects between the designed wavefunctions of strongly correlated electrons.

In addition, our observations also offer valuable insight into the quest for novel quantum phenomena using oxide heterostructures, since the U/W ratio can be controlled by designing the wavefunction of their strongly correlated electrons. Thus, we believe that the successful control of MIT has a significant impact on the field.

Also, the authors have only shown this resonant tunneling for one particular combination of the QW thickness V_2TLV_6 . The authors should show at least one different thickness configuration of the QW for the same SVO/STO/SVO system or use different materials and observe the same phenomena to say that this can be universally seen in oxide materials.

Reply: To meet the reviewer's requirement, we have performed additional experiments on V_2TLV_5 and added the results to the Supplementary Information (**Supplementary Note 11. RT-driven MIT in V_2TLV_5 double QW structures**). It is evident that identical phenomena occur in different thickness configurations of the double QW, although the intensity of the coherent band is weaker in V_2TLV_5 owing to the slight offset of the corresponding quantization energies (Fig. S1). The additional data demonstrate that the RT-induced MIT can be universally observed in the oxide QWs, and this strengthens the validity of our conclusions.

2. Related to the above comment, why does the STO thickness need to be 2 ML, not 1 or 3? Intuitively, I would expect that 1 ML STO as the interval layer should also work. What determines the length scale of the insulating layer needed for the resonant tunneling?

Reply: The reason for choosing a thickness of 2 ML for the STO barrier layer is to completely isolate the two QWs for demonstrating the occurrence of the RT effects. In general, it is difficult to obtain a rigorous single-layer growth by monitoring the RHEED oscillations, since the oscillation period is unknown during the first oscillation. Thus, there is a possibility that a 1-ML thick barrier could present patches of 0-ML growth, which could lead to direct contact between the two QWs through the leaky 0-ML areas of the STO barrier layer. Although the contribution of this direct contact is expected to be small enough, we utilized the 2 ML STO barrier layer to make extra sure of the occurrence of the RT effects.

As Reviewer #3 pointed out, the 1-ML STO should also work as the interval layer. To demonstrate this, we performed additional experiments for $L = 1$ and 3 in $V_2T_LV_6$ structures and added the results in the Supplementary Information (Supplementary Note 10. Series of ARPES images for $V_2T_LV_6$ double QW structures with reducing L from ∞ to 1). As the reviewer expected, the interval layers of 1-ML and 3-ML STO also work (Fig. S17). Furthermore, additional DFT calculations for $V_2T_LV_6$ (Fig. S25) also support that the same RT occurs in 1 ML and 3 ML as in 2 ML. We appreciate the reviewer's comment, because it initiated an additional verification of our interpretation of the RT effects.

Figure S18 shows the plot of the quasiparticle peak (MDC peak) intensity at E_F as a function of L . A steep attenuation of the intensity with increasing L is observed, reflecting the tunneling effect. By fitting the data with the exponential decay function of $\exp(-L/\lambda)$, we determined the length scale of the insulating layer needed for the RT-induced MIT to be $\lambda = 0.59$ nm (~ 1.5 ML). The characteristic length scale is in line with the DFT results (Fig. S26).

In conventional tunneling phenomena for a rectangular potential, the tunneling probability T is described by the following equation:

$$T = \exp \left[-2\sqrt{(2m^*/\hbar^2)(E_B - E)}L_B \right].$$

Here, m^* is the effective mass of the electron, \hbar is the Plank constant, E_B is the potential barrier height, E is the energy of the electron, and L_B is the width of the potential barrier. Note that in the present double QW structures only electrons in the specific QW states can penetrate to the neighboring QW through energetically close QW states (resonant states); the existence probability of electrons in the other QW states decays steeply owing to the absence of the corresponding QW states. Therefore, in the first approximation, the length scale is determined by the potential height and width of the barrier layer. Indeed, the exponential decay of the quasiparticle peak intensity strongly suggests the tunneling nature of the observed MIT, although we should consider the fact that the plot in Fig. S18 is the quasiparticle intensity of the top QW, rather than the tunneling probability itself.

The MIT of strongly correlated materials originates from the delicate balance between U and W . In the RT-driven MIT, U varies through the propagation of electrons between two QWs, and resultant change in the U/W ratio induces the MIT. Thus, the complicated interactions of electrons in double QW structures should be considered. However, it is far beyond the scope of this study to clarify them. We would like to leave this as a future theoretical work. We have commented on this issue in the manuscript (lines 198–204).

3. The authors have only showed the ARPES data taken at $h\nu=88\text{eV}$. Since they are doing the experiments at synchrotron radiation, why not show the data taken at other photon energies? They claim that “It should be noted that the observed metallic states are at the top 2-ML SVO layer, since the mean-free-path of the photoelectrons in the present experimental condition is about 0.4–0.6 nm (corresponding to 1.0–1.5 ML in the present case) (refs 24–26).” in lines 114-116. But shouldn't they seem some difference by using the photon energy of 30-2000eV that is available in this beamline?

Reply: Reviewer 3 suggests showing ARPES data taken at different photon energies. Following the reviewer's suggestion, we have added the data to the Supplementary Information (Supplementary Note 12. Contributions from bottom QW states in ARPES images). Figure S21 shows a series of ARPES images taken at different photon energies in the range of 88–1013 eV. As can be seen in Fig. S21, there are no fundamental differences in the observed band structure under the change in photon energies, except for the peak broadening attributable to the poor energy resolution at higher photon energies. These results indicate that the observed metallic states are at the top 2-ML SVO layer.

We guess that the reviewer is concerned about the contribution from the bottom metallic QW states. To address this, we performed an additional ARPES experiment on the 4-ML STO/ 6-ML SVO bilayer structure. Using the bilayer, we can estimate the contribution from the bottom metallic QW states to the ARPES results for 2-ML SVO/ 2-ML STO/ 6-ML SVO double QW structures. Figure S22 presents a comparison of the ARPES images at the same intensity scale. From the additional ARPES data, it is clear that the contribution from buried metallic QW states is negligible in the present ARPES data taken at $h\nu = 88\text{ eV}$ (Figs. 2 and 3).

Meanwhile, we observed very weak but distinct QW states of the buried QW in the 4-ML STO/ 6-ML SVO bilayer structure when the dynamic range of the ARPES image was expanded (x20). We discuss this in the response to Comment 5 below.

4. The authors compare the present results with the field-effect transistors of semiconductors. I would like to actually see the transport data of the present systems (the difference of the conductivity between metallic state V2T2V6 and marginal Mott state V2T10V6).

Reply: We had attempted to measure the transport properties of double QW structures. Unfortunately, we were unable to measure the conductivity of the upper QW layer. This is because 1) the surface of SVO is unstable in the atmosphere, and 2) STO barrier layers and STO substrates are reduced and become conductive during

the growth process. For these reasons, we have employed photoemission spectroscopy to verify the concept of RT-induced MIT.

In future, we intend to fabricate double QW structures using materials besides STO to determine the conductivity of the marginal Mott state and metallic states. However, these transport measurements are far beyond the scope of this study. Therefore, we would like to leave it for future research.

5. In Fig. 3, the authors compare the data of vacuum / SVO 6ML / STO 2ML/ SVO 2ML /STO substrate and vacuum / SVO 2ML / STO 2ML/ SVO 6ML /STO substrate (although the vacuum and STO substrate are not explicitly written). Are these two comparable? Since the substrate and vacuum should not have the same contribution in the confinement of the QW, I wonder if this comparison is really reasonable or not. Please make some comments on this point.

Reply: Reviewer 3 is concerned about the difference in the confinement of QW states between the vacuum and STO. Since the substrate and vacuum should have different contributions to the confinement of the QW states, it is important to determine whether this difference is negligible. We have addressed this issue already [Ref. 51: K. Yoshimatsu *et al.*, Phys. Rev. B **88**, 115308 (2013)]. In Ref. 51, we investigated the QW states of vacuum/SVO/STO and STO/SVO/STO QW structures and found that there was little change in the quantization levels below 200 meV for both QW structures. A detailed analysis based on the phase-shift quantization rule reveals that an almost ideal quantum confinement is achieved for both the vacuum/SVO and STO/SVO interfaces, demonstrating that the vacuum and STO contribute almost identically to the confinement of the QW states of the SVO layer.

This fact is further confirmed by the comparison of QW states between the vacuum/ 6-ML SVO/ STO sub. and 4-ML STO/ 6-ML SVO/ STO sub. as shown in Fig. S23 (the same as in Fig. S22, but with the dynamic range expanded by 20 times for Fig. S22b). It is evident that there was little change in the quantization levels of the QW states. It should be noted that the invariance of the QW states at the surface (vacuum/SVO interface) and the interface (STO/SVO) provides further evidence for the formation of a chemically abrupt interface between STO and SVO.

Based on these ARPES studies, we conclude that the comparison is reasonable. We have added some explanations to the Supplementary Information (Supplementary Note 13. Quantum confinement at vacuum/SrVO₃ and SrTiO₃/SrVO₃ interfaces).

In summary, I do not see significant novel findings in the present manuscript for publication in Nat. Comm. at this stage. After the authors address the above comments properly and make modifications, I would be happy to review it again.

These additional data and analyses indisputably demonstrate our most interesting finding: the occurrence of MIT by resonant tunneling between the two QWs. We have revised the manuscript according to the reviewer's useful suggestions, including the results of the additional experiments.

We believe that the concerns of Reviewer #3 have been sufficiently addressed in the revised manuscript. This study provides a significant step towards controlling the MIT, as well as novel quantum phenomena, of strongly correlated oxides through designing wavefunctions of strongly correlated electrons. Thus, we believe that the revised manuscript appeals to a broader audience. We wish to highlight that Reviewers #1 and #2 highly appreciate the importance of the present study as well as the technical quality.

REVIEWERS' COMMENTS

Reviewer #1 (Remarks to the Author):

The authors have made proper responses to my comments. Thus I would like to recommend its publication in the current form.

Reviewer #3 (Remarks to the Author):

The manuscript of Yukawa et al has been revised nicely according to the referee reports. Though I still hope the authors can perform transport measurements of the present samples, (the difference of the conductivity between metallic state V2T2V6 and marginal Mott state V2T10V6), I agree that this may be difficult due to the fact that SVO is unstable

in the atmosphere. I hope the authors know that there are in situ transport measurement systems around the world that are capable to measure the intrinsic transport of materials that are not stable in air. Hopefully, their present finding with ARPES can be checked with transport measurements in the near future. Thus I recommend the present manuscript to be published in Nature Communications.

Reviewer #2 (Remarks to the Author):

The authors have taken the previous feedback seriously and addressed most of my concerns. I request one more dataset (which the authors already possess) in relation to comment #1, which should be simple enough to do. Good luck.

Response to comment #1

The first step of this interesting core-level analysis (Fig. S4) shows that the surface of a Nb:STO substrate is not likely damaged by deposition of SVO. However, single crystals are more robust to damage compared to ultrathin (STO) epitaxial layers.

As such, I ask that the authors enlarge and fit representative segments of the Ti 2p, Sr 3d and V 2p from Fig. S5, and via comparison show that they do they remain identical at 2-3 stages of the process. For example: by plotting a normalized version of V 2p with L=0, 4 and 10 (and repeating for Ti 2p and Sr 3d). Since the authors already have this data, I expect this to be a simple task, and the results would be most convincing (and useful on their own).

The authors may wish to discuss their finding in comparison to recent interesting reports on core levels of SVO (that were exposed to the atmosphere) [e.g. Bourlier et al., (2021)

<https://doi.org/10.1016/j.apsusc.2021.149536> , Fouchet et al., (2018)

<https://doi.org/10.1063/1.4998004>] (this last paragraph is only a suggestion)

Comment #3B

This is a nice schematic and response. I would suggest (not mandatory) to include schematic contacts (ground and gate), and perhaps to add the material names to the QW and barrier, if it doesn't overload the illustration. This is merely a suggestion.

Reply to Reviewers

(referee's comments in blue and our responses in black)

Reply to Reviewer #1

The authors have made proper responses to my comments. Thus, I would like to recommend its publication in the current form.

Reply: We are grateful to Reviewer #1 for the strong recommendation for the publication of this manuscript in Nat. Commun. We sincerely thank the reviewer for his/her valuable suggestions to improve our manuscript through the review.

Reply to Reviewer #2 (Prof. Lior Kornblum):

The authors have taken the previous feedback seriously and addressed most of my concerns. I request one more dataset (which the authors already possess) in relation to comment #1, which should be simple enough to do. Good luck.

Reply: We are very grateful to Prof. Kornblum for the strong recommendation for the publication of this manuscript in Nat. Commun. We sincerely thank the reviewer for his/her valuable suggestions to improve our manuscript through the review. We address the minor points raised by the reviewer below.

Response to comment #1

The first step of this interesting core-level analysis (Fig. S4) shows that the surface of a Nb:STO substrate is not likely damaged by deposition of SVO. However, single crystals are more robust to damage compared to ultrathin (STO) epitaxial layers.

As such, I ask that the authors enlarge and fit representative segments of the Ti 2p, Sr 3d and V 2p from Fig. S5, and via comparison show that they do they remain identical at 2-3 stages of the process. For example: by plotting a normalized version of V 2p with L=0, 4 and 10 (and repeating for Ti 2p and Sr 3d). Since the authors already have this data, I expect this to be a simple task, and the results would be most convincing (and useful on their own).

The authors may wish to discuss their finding in comparison to recent interesting reports on core levels of SVO (that were exposed to the atmosphere) [e.g. Bourlier et al., (2021)

<https://doi.org/10.1016/j.apsusc.2021.149536>, Fouchet et al., (2018) <https://doi.org/10.1063/1.4998004>
(this last paragraph is only a suggestion)

Reply: In accordance with his valuable suggestions for the improvement of our manuscript, we have added the requested dataset in Supplemental information (Supplementary Note 5-3: Chemical analysis for core-level spectra). Also, we have added some discussion in Supplemental information by citing these works, together with the related works (Supplementary Refs. 11-17 in the revised Supplementary Information).

The invariance of the chemical environments at the interface to SrVO₃ (SVO) between single crystalline SrTiO₃ (STO) substrates and epitaxial STO barrier layers are demonstrated in Fig. R1, where a Ti 2*p* core level for SVO (2 ML)/ STO substrates ($L = \infty$) is compared with that of SVO (2 ML)/STO (10 ML)/SVO (6 ML) structures ($L = 10: V_2T_{10}V_6$). The line shapes of the Ti 2*p* core levels are identical, demonstrating the invariance of the chemical environments between the two.

FIG R1: The comparison of Ti 2*p* core levels between single crystalline STO substrates [SVO (2 ML)/ STO substrates] and epitaxial STO barrier layers [SVO (2 ML)/STO (10 ML)/SVO (6 ML) structures].

Comment #3B

This is a nice schematic and response. I would suggest (not mandatory) to include schematic contacts (ground and gate), and perhaps to add the material names to the QW and barrier, if it doesn't overload the illustration. This is merely a suggestion.

Reply: In accordance with his valuable suggestions, we have included schematic contacts (ground and gate) and added the notations in Fig. 4.

Reply to Reviewer #3

The manuscript of Yukawa et al has been revised nicely according to the referee reports. Though I still hope the authors can perform transport measurements of the present samples, (the difference of the conductivity between metallic state V2T2V6 and marginal Mott state V2T10V6), I agree that this may be difficult due to the fact that SVO is unstable in the atmosphere. I hope the authors know that there are in situ transport measurement systems around the world that are capable to measure the intrinsic transport of materials that are not stable in air. Hopefully, their present finding with ARPES can be checked with transport measurements in the near future. Thus I recommend the present manuscript to be published in Nature Communications.

Reply: We are grateful to Reviewer #3 for the strong recommendation for the publication of this manuscript in Nat. Commun. We sincerely thank the reviewer for his/her valuable suggestions to improve our manuscript through the review.

According to the reviewer's valuable advice, we will measure the conductivity of the marginal Mott state and metallic states by in situ transport measurement systems in the near future.